# Changes in the Chemical Composition of Six Lettuce Cultivars (*Lactuca sativa* L.) in Response to Biofortification with Iodine and Selenium Combined with Salicylic Acid Application

**Sylwester Smoleń [1],\*** , **Iwona Kowalska [1]** , **Peter Kováčik [2]** , **Włodzimierz Sady [1]** , **Marlena Grzanka [1]** and **Umit Baris Kutman [3]**

1   Unit of Plant Nutrition, Institute of Plant Biology and Biotechnology, Faculty of Biotechnology and Horticulture, University of Agriculture in Krakow, 31-425 Kraków, Poland; iwona.kowalska@urk.edu.pl (I.K.); wlodzimierz.sady@urk.edu.pl (W.S.); marlena.grzanka14@gmail.com (M.G.)
2   Department of Agrochemistry and Plant Nutrition, Slovak University of Agriculture in Nitra, Tr. A. Hlinku 2, 949 01 Nitra, Slovakia; peter.kovacik@uniag.sk
3   Department of Bioengineering, Faculty of Engineering and Architecture, Konya Food and Agriculture University, Beysehir Street, 42080, Konya, Turkey; bkutman@gtu.edu.tr
\*   Correspondence: sylwester.smolen@urk.edu.pl; Tel.: +48-12-662-52-39

**Abstract:** A two-year greenhouse study was conducted to assess the effects of the application of I (as $KIO_3$), Se (as $Na_2SeO_3$), and salicylic acid (SA) in nutrient solutions on the chemical composition of six lettuce cultivars, i.e., two butterhead lettuces (BUTL), "Cud Voorburgu" and "Zimująca"; two iceberg lettuces (ICEL), "Maugli" and "Królowa lata"; and two *Lactuca sativa* L. var. *crispa* L. (REDL) cultivars, "Lollorossa" and "Redin", grown in the NFT (nutrient film technique) system. The treatments were as follows: control, I+Se, I+Se+0.1 mg SA $dm^{-3}$, I+Se+1.0 mg SA $dm^{-3}$, and I+Se+10.0 mg SA $dm^{-3}$. $KIO_3$ was used at a dose of 5 mg I $dm^{-3}$, while $Na_2SeO_3$ was used at 0.5 mg Se $dm^{-3}$. The application of I+Se was a mild abiotic stress factor for the plants of the ICEL and REDL cultivars. In contrast, I+Se did not have a negative impact on the BUTLcultivars. The application of 1.0 mg SA $dm^{-3}$ improved the biomass productivity in all cultivars compared with I+Se. In the majority of the cultivars, the applied combinations of I+Se and I+Se+SA resulted in a reduction in the nitrate(V) content that was beneficial to the consumer and increased levels of sugars, phenols, phenylpropanoids, flavonols, and anthocyanins. In addition, an increase in ascorbic acid content was observed, but only in the BUTL cultivars and REDL "Redin". The application of I, Se, and SA had upward or downward effects on the concentrations of N, K, P, Ca, Mg, S, Na, B, Cu, Fe, Mn, Mo, and Zn in the leaves.

**Keywords:** beneficial elements; sugars; phenolic compounds

## 1. Introduction

The last 10 years saw many studies oriented toward the biofortification (enrichment) of crop plants with various elements. This is aimed at reducing the deficit of minerals in the human diet and animal feed, as this problem is widespread throughout the world [1,2].

It was repeatedly demonstrated that the use of agrotechnical means of enriching plants with minerals is an optimal and low-cost method of introducing them into the trophic chain, including the human diet [3,4].

It is estimated that iodine (I) and selenium (Se) deficiencies affect approximately 30% and 15% of the world's population, respectively [5]. For this reason, many research centers around the world conduct research toward the biofortification of plants with I [6–8] or Se [9–12].

I and Se are included in the group of "elements beneficial to plants" [13–15]. The problem with the simultaneous enrichment of plants with both these elements is a relatively new research issue. Apart from the authors' scientific center (citations of our own research are omitted), similar research was conducted by, for example, Zhu et al. [16] on spinach and Mao et al. [17] on the cultivation of wheat, maize, soybean, potato, canola, and cabbage bark. Those species were mainly studied to determine the effectiveness of enriching them with I and Se. There were very few studies, however, that determined the chemical compositions of plants or their physiological states resulting from combined application of I and Se. This type of research was conducted on buckwheat and pumpkin sprouts [18], pea [19], and kohlrabi sprouts [20], among others.

The hydroponic NFT (nutrient film technique) system was used for many years for research in the field of mineral nutrition of plants. The NFT technique allows model studies on the functioning of mineral metabolism in plants and also on the biochemical and physiological processes occurring in them to be carried out [21].

Salicylic acid (SA) is involved in the resistance response of plants to abiotic and biotic (including pathogens) stress factors [22]. The introduction of salicylic acid into nutrient solutions in hydroponic cultures may increase plant resistance to these stress factors. The application of SA to nutrient solutions may also increase the uptake of minerals, including iodine, by plants [23].

To date, no studies were conducted to determine the effect of combined biofortification with I and Se on the chemical compositions of various botanical varieties of lettuce. The research results presented here fill the information gap in this area of knowledge.

The aim of the study was to determine how the process of biofortification of different lettuce cultivars with I and Se in the presence of SA (in a hydroponic NFT system) affects the parameters of plant chemical composition that are important to the consumer, including the levels of selected nutraceuticals.

## 2. Materials and Methods

### 2.1. Plant Material and Treatments

A two-year study was conducted in a greenhouse with hydroponic cultivation of six lettuce cultivars in an NFT system: two butterhead lettuces (BUTL), "Cud Voorburgu" and "Zimująca"; two iceberg lettuces (ICEL), "Maugli" and "Królowa lata" (all these four cultivars are classified as *Lactuca sativa* L. var. *capitata*); and two *Lactuca sativa* L. var. *crispa* L. cultivars (REDL), "Lollorossa" and "Redin", which have little red-colored leaves.

The experiment was conducted in a greenhouse of the Faculty of Biotechnology and Horticulture, University of Agriculture, in Kraków. Each year, seeds were sown into rockwool plugs (Grodan, Rockwool B.V., Roermond, the Netherlands) at the end of August and the beginning of September. Seedlings at the two-leaf stage were placed into holes (spaced 25 cm apart) in Styrofoam slabs filling NFT beds (the dry hydroponic method). No additional substrate was used. The greenhouse was equipped with five individual NFT sets with 1.300-dm$^3$ containers of nutrient solution, facilitating lettuce cultivation in recirculating hydroponics.

After the seedlings were planted in the hydroponic system, day and night temperatures were set to 15 and 10 °C, respectively. From the beginning of October to the end of the experiment, natural light was supplemented with the use of 600-W high-pressure sodium lamps between 5:00 and 10:00 a.m.

The study included the application of I (as KIO$_3$, p.a., Avantor Performance Materials, Gliwice, Poland), Se (as Na$_2$SeO$_3$, puriss. p.a., Sigma-Aldrich, St. Louis, MO, USA), and SA (puriss. p.a., Avantor Performance Materials) into nutrient solutions for the cultivation of all types of cultivars. The tested treatments were as follows: (1) control (with trace I and Se levels in the nutrient solution from the applied fertilizers; approximately 30 µg·dm$^{-3}$ I and 8.5 µg dm$^{-3}$ Se), (2) I + Se, (3) I + Se

+ 0.1 SA (0.1 mg SA dm$^{-3}$ nutrient solution, i.e., 0.724 μM SA), (4) I + Se + 1.0 SA (1.0 mg SA dm$^{-3}$ nutrient solution, i.e., 7.24 μM SA) and (5) I + Se + 10.0 SA (10.0 mg SA dm$^{-3}$ nutrient solution, i.e., 72.4 μM SA). KIO$_3$ was used at a dose of 5 mg I dm$^{-3}$ (i.e., 39.4 μM I), while Na$_2$SeO$_3$ was used at 0.5 mg Se dm$^{-3}$ (i.e., 6.3 μM Se). I, Se, and SA were instantly introduced into the nutrient solutions beginning at the three-to four-leaf stage (formation of the rosette). The experiment was conducted according to a randomized block design with four replications—three plants per replicate in each treatment. The plants were grown in a nutrient solution at pH 5.50 and EC (electrical conductivity) 1.8 mS·cm$^{-1}$ with the following concentrations of macro- and micronutrients (mg·dm$^{-3}$): 120 N, 40 P, 170 K, 35 Mg, 150 Ca, 1.5 Fe, 0.55 Mn, 0.25 Zn, 0.2 B, 0.09 Cu, and 0.04 Mo. These concentrations were equivalent to 8.57 mM N, 1.29 mM P, 4.35 mM K, 1.44 mM Mg, 3.74 mM Ca, 26.9 μM Fe, 10.0 μM Mn, 3.8 μM Zn, 18.5 μM B, 1.4 μM Cu, and 0.4 μM Mo.

For each treatment, 1.300 dm$^3$ of each nutrient solution was stored in a separate container and periodically administered to the cultivation slabs. The frequency of watering was adjusted for the growth stage of lettuce and the weather conditions. Plants were cultivated in the system of recirculating nutrient solution without a disinfection system. The plants used the same nutrient solutions throughout the entire period.

Lettuce harvest was conducted at the beginning of December in each year of the study.

### 2.2. Plant Analysis

For the analyses described in Sections 2.2.1 and 2.2.2, lettuce heads were cut in half and mixed in order to obtain a representative sample of all leaves (old and young) from all heads in each treatment.

The sections that follow provide a general description of the analysis methods using the Beckman Coulter PA 800 Plus capillary electrophoresis system (to estimate l-ascorbic acid and sugars) and an ICP-OES (inductively coupled plasmaoptical emission spectrometer; Prodigy Spectrometer Leeman Labs, New Hampshire, MA, USA) to estimate K, P, Ca, Mg, S, Na, B, Cu, Fe, Mn, Mo, and Zn levels. A more detailed description of the method and conditions used for performing these analyses was presented in our previous publication [23].

### 2.2.1. Analysis of Fresh Lettuce Heads (Leaves)

Fresh lettuce leaves were analyzed to determine the concentrations of l-ascorbic acid, sugars, phenols, phenylpropanoids, flavonols, anthocyanins, and nitrates(V).

The concentration of l-ascorbic acid in the leaves was analyzed by capillary electrophoresis after homogenization of 20-g samples in 80 cm$^3$ of 2% oxalate acid (puriss. p.a., Avantor Performance Materials), and then further centrifugation at 4500 rpm for 15 min at 5 °C was carried out. The supernatants were filtered through a 0.25-μm cellulose acetate membrane filter and analyzed using the capillary electrophoresis method with DAD (diode array detector) detection.

In order to determine the concentrations of sugars and phenolic compounds, fresh leaves were extracted with boiling ethanol (96%, Destylernia "Polmos" Sp. z o.o., Kraków, Poland) using a reflux condenser. The levels of fructose, glucose, and sucrose (and their sum as total sugars) were estimated by the capillary electrophoresis method with DAD detection. The concentrations of phenols, phenylpropanoids, flavonols, and anthocyanins were determined spectrophotometrically after the samples reacted with 0.1% HCl (puriss. p.a., Avantor Performance Materials) dissolved in ethanol [24].

The level of nitrates(V) was estimated usingthe FIA (flow injection analysis) technique with the use of an FIA Modula MLE (Germany) after homogenization of 5-g samples in 100 cm$^3$ of 2% acetic acid (puriss. p.a., Avantor Performance Materials) [25].

### 2.2.2. Analysis after Sample Drying

Fresh lettuce leaves (after washing in distilled water) were dried at 7 °C in a laboratory dryer with forced air circulation and then ground in a FRITSCH Pulverisette 14 variable speed rotor mill (Idar-Oberstein, Germany) using a 0.5-mm sieve. Samples were subsequently analyzed to determine

the concentrations of the following elements: N using the Kjeldahl method, and K, P, Ca, Mg, S, Na, B, Cu, Fe, Mn, Mo, and Zn (only in the leaves) using the ICP-OES technique.

### 2.3. Data Analysis

The data were subjected to variance analysis using the two-way (treatment × cultivar) analysis of variance (ANOVA) module of Statistica 12.0 PL software (https://www.tibco.com/products/data-science, StatSoft Inc., Tulsa, OK 74104, USA). We decided to verify whether the tested factors had significant influences on lettuce biomass, as well as on the analyzed parameters of the chemical composition of the plants. The LSD (least significant difference) test was used to determine the significance between the means at a significance level of $p < 0.05$. Numerical values of LSD were calculated for the Duncan test using statistical tables. Those calculations were performed manually after the ANOVA analysis with Statistica 12.0 PL software.

The index of tolerance (TI) was also calculated. Firstly, the weight (biomass) of lettuce heads was measured as a parameter for the calculation of the TI toward iodine. Then, the TI was calculated as the weight of lettuce heads after treatment with the tested compounds divided by the weight of lettuce heads of the control plants and expressed as a percentage of the control [26]: TI = (average weight of lettuce heads treated with tested compounds/average weight of control lettuce heads) × 100%. TI values below 100% indicated a lower level of plant biomass relative to the control.

## 3. Results

### 3.1. TI and Information on the Yield and Biofortification Effect

The TI calculations showed that the introduction of I+Se (without SA) into the nutrient solution resulted in the weakening of the growth of heads of four lettuce cultivars: ICEL "Maugli" and "Królowa lata", as well as REDL "Lollorossa" and "Redin"; the latter proved to be the most sensitive cultivar to I and Se (Figure 1). In contrast, the application of I+Se did not have a negative impact on the BUTL cultivars "Cud Voorburgu" and "Zimująca" (TI > 100%).

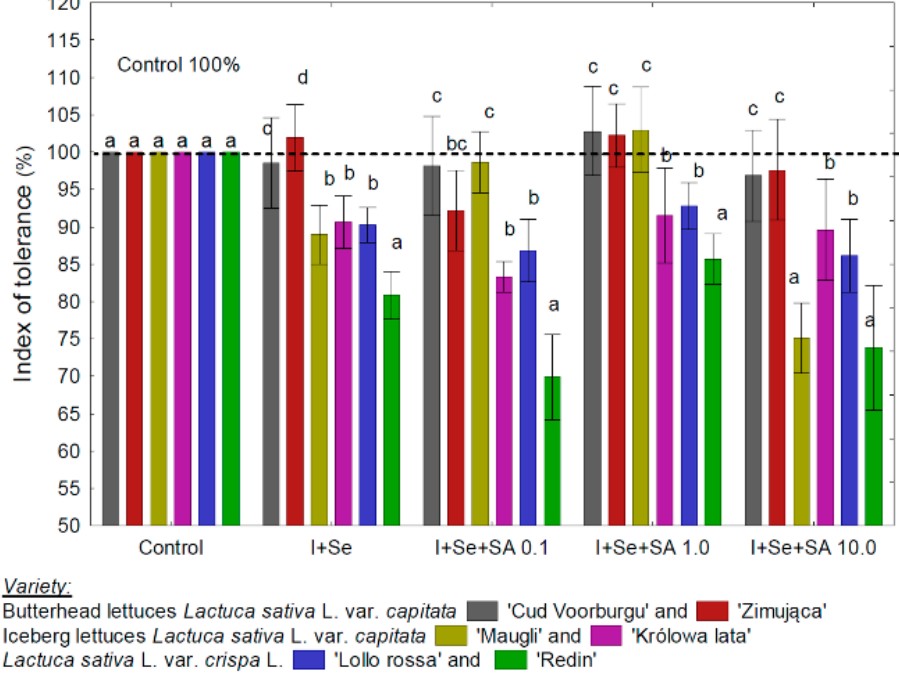

**Figure 1.** Index of tolerance. Means for cultivars followed by the same letters separately for each treatment (type of nutrient solution) are not significantly different at $p < 0.05$. Bars indicate standard errors ($n = 8$).

The additional enrichment of nutrient solutions with SA at 1.0 mg SA dm$^{-3}$ improved the growth of heads of all the tested lettuce cultivars in relation to the plants cultivated on the medium containing only I+Se (Figure 1). Meanwhile, SA applied at the lowest and highest doses, i.e., 0.1 and 10.0 mg SA dm$^{-3}$, weakened the growth of lettuce heads in comparison with fertilization with I and Se (without SA).

All the cultivars of lettuce were also analyzed for the following: I, Se, selenomethionine (SeMet), and selenocysteine (SeCys). The results of those analyses are presented in a separate publication [27]. The level of I accumulation formed the following order (iodine content in mg Ikg$^{-1}$dry weight (d.w.) only for the I+Se treatment): REDL "Lollorossa" (292.3) > BUTL "Cud Voorburgu" (252.4) > REDL "Redin" (245.0) > BUTL "Zimująca" (206.2) > ICEL "Królowa lata" (123.1) > ICEL "Maugli" (114.8). The order of Se content of the leaves was as follows (selenium content in mg Sekg$^{-1}$d.w. only for the I+Se treatment): BUTL 'Cud Voorburgu' (10.8) > BUTL "Zimująca" (9.7) ≥ REDL "Lollorossa" (9.4) ≥ REDL "Redin" (9.3) > ICEL "Maugli" (8.2) > ICEL "Królowa lata" (7.8). The content of SeMet (in mg SeMetkg$^{-1}$d.w.) in lettuce for the I+Se treatment was in the range from 2.65 in REDL "Lollorossa" to 5.36 in ICEL "Królowa lata"; moreover, content of SeCys (in mg SeCyskg$^{-1}$d.w.) was in the range from 0.60 in REDL "Redin" to 2.07 BUTL "Cud Voorburgu". Those results are referred to in the discussion section, only to the extent necessary to explain the causes of changes in the chemical composition of lettuce following the application of I, Se, and SA.

## 3.2. Ascorbic Acid and Nitrates(V)

The individual lettuce cultivars showed statistically significant and different responses to the application of I+Se and SA to the nutrient solution in terms of concentrations of ascorbic acid and nitrate(V) ($NO_3^-$) (Figure 2A,B).

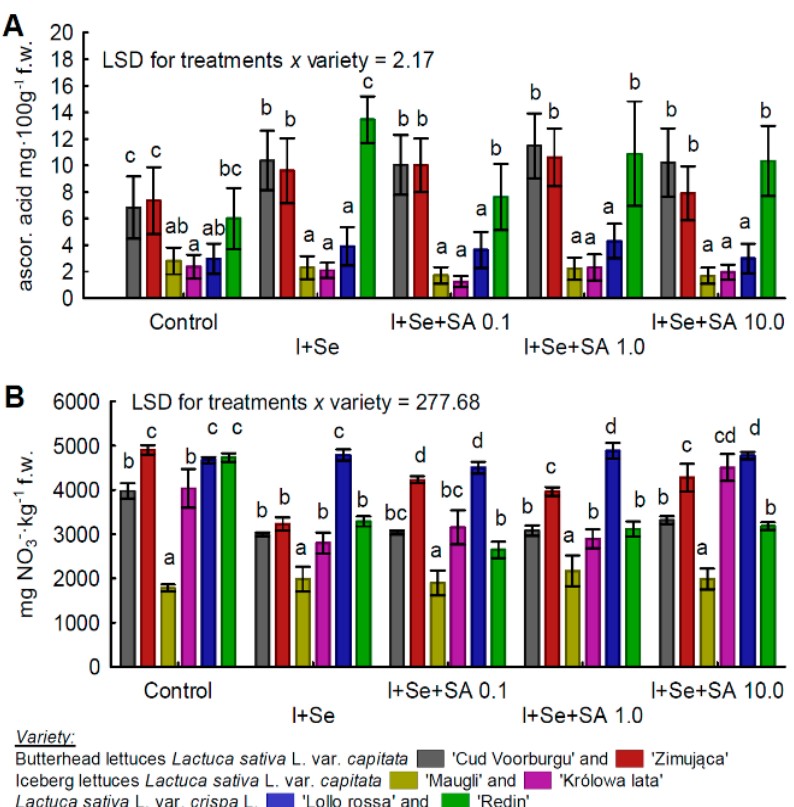

**Figure 2.** Concentrations of ascorbic acid (**A**) and nitrate(V) ($NO_3^-$) (**B**) in lettuce. Means for cultivars followed by the same letters separately for each treatment (type of nutrient solution) are not significantly different at *p*< 0.05. LSD (least significant difference) for interaction "treatments × variety" at *p*< 0.05. Bars indicate standard errors (*n* = 8). f.w.—fresh matter weight.

Compared with the control, after the application of I+Se and SA, the ascorbic acid content of lettuce was significantly increased only in the BUTL cultivar "Cud Voorburgu" and in the REDL cultivar "Redin" (Figure 2A). In the remaining cultivars, the ascorbic acid content was the same as in the control.

The highest ascorbic acid content was shown by the BUTL cultivars "Cud Voorburgu" and "Zimująca", and the lowest was shown by ICEL "Królowa lata" and "Maugli" (Table 1).

The tested nutrient solutions with the addition of I+Se and SA (relative to the control) caused, to the same extent, a significant reduction in the levels of nitrates(V) in the heads of the following lettuce cultivars: BUTL "Cud Voorburgu" and "Zimująca",as well as REDL "Redin" (Figure 2B). In the case of the ICEL cultivar "Królowa lata", the application of I+Se and SA at doses of 0.1 and 1.0 mg·dm$^{-3}$ resulted in a significant reduction in the nitrate(V) content of lettuce, whereas the highest dose of 10 mg SA dm$^{-3}$ caused a significant increase. On the other hand, in the ICEL cultivar "Maugli" and the REDL cultivar "Lollorossa", the application of I, Se and, additionally, SA did not cause significant changes in the nitrate(V) content of lettuce compared with the control. The highest level of accumulation of nitrates(V) was found in the REDL cultivar "Lollorossa", and the lowest was found in the ICEL cultivar "Maugli" (Table 1).

**Table 1.** The ordering of the lettuce varieties in terms of the level of the content of the compounds and the elements in the leaves (heads). BUTL—butterhead lettuce; ICEL—iceberg lettuce; REDL—*Lactuca sativa* L. var. *crispa* L. (red-leaved lettuce).

| Compounds/Elements | Lettuce varieties ordered from the highest to the lowest content of individual compounds and elements in the leaves (heads) ($n = 8$). |
|---|---|
| Ascorbic acid | **BUTL** "Zimująca" = **BUTL** "Cud Voorburgu" = **REDL** "Redin" >**REDL** "Lollorossa" >**ICEL** "Królowa lata" = "**ICEL** "Maugli" |
| Nitrate(V) | **REDL** "Lollorossa" >**BUTL** "Zimująca" >**ICEL** "Królowa lata" = **REDL** "Redin" >**BUTL** "Cud Voorburgu" >**ICEL** "Maugli". |
| Glucose, fructose, and total sugars | **ICEL** "Maugli" >**BUTL** "Zimująca" **BUTL** "Cud Voorburgu" = **ICEL** "Królowa lata" >**REDL** "Redin" > **REDL** "Lollorossa" |
| Sucrose only | **BUTL** "Zimująca" >**BUTL** "Cud Voorburgu" >**REDL** "Redin" = **REDL** "Lollorossa" >**ICEL** "Maugli" = **ICEL** "Królowa lata" |
| Phenols, phenylpropanoids, flavonols, and anthocyanins | **BUTL** "Zimująca" >**REDL** "Redin" >**BUTL** "Cud Voorburgu" >**REDL** "Lollorossa" >**ICEL** "Królowa lata" >**ICEL** "Maugli". |
| N | **REDL** "Redin" >**BUTL** "Cud Voorburgu" >**BUTL** "Zimująca" = **ICEL** "Maugli" = **ICEL** "Królowa lata" >**REDL** "Lollorossa" |
| P | **BUTL** "Cud Voorburgu" >**REDL** "Redin" >**REDL** "Lollorossa" >**ICEL** "Maugli" >**ICEL** "Królowa lata" = **BUTL** "Zimująca" |
| K | **REDL** "Lollorossa" >**ICEL** "Królowa lata" >**BUTL** "Zimująca" = **REDL** "Redin" >**BUTL** "Cud Voorburgu" = **ICEL** "Maugli" |
| Mg | **REDL** "Lollorossa" >**BUTL** "Cud Voorburgu" >**BUTL** "Zimująca" >**ICEL** "Królowa lata" >**REDL** "Redin" = **ICEL** "Maugli" |
| Ca | **REDL** "Lollorossa" >**BUTL** "Cud Voorburgu" > **BUTL** "Zimująca" >**ICEL** "Królowa lata" >**REDL** "Redin" >**ICEL** "Maugli" |
| S | **ICEL** "Królowa lata" >**REDL** "Redin" = **ICEL** "Maugli" >**BUTL** "Cud Voorburgu" = **BUTL** "Zimująca" >**REDL** "Lollorossa" |
| Na | **REDL** "Lollorossa" >**BUTL** "Cud Voorburgu" >**ICEL** "Królowa lata" = **BUTL** "Zimująca" > **REDL** "Redin" >**ICEL** "Maugli" |
| B | **REDL** "Lollorossa" > **REDL** "Redin" > **BUTL** "Zimująca" >**BUTL** "Cud Voorburgu" = **ICEL** "Królowa lata" >**ICEL** "Maugli" |
| Cu | **BUTL** "Cud Voorburgu" >**BUTL** "Zimująca" >**REDL** "Lollorossa" >**ICEL** "Maugli" = **REDL** "Redin" >**ICEL** "Królowa lata" |
| Fe | **REDL** "Redin" > **BUTL** "Cud Voorburgu" >**REDL** "Lollorossa" = **BUTL** "Zimująca" > **ICEL** "Królowa lata" >**ICEL** "Maugli" |
| Mn | **REDL** "Lollorossa" >**ICEL** "Maugli" >**ICEL** "Królowa lata" > **REDL** "Redin" = **BUTL** "Zimująca" > **BUTL** "Cud Voorburgu" |
| Mo | **REDL** "Redin" = **REDL** "Lollorossa" >**ICEL** "Maugli" = **BUTL** "Zimująca"= **ICEL** "Królowa lata" >**BUTL** "Cud Voorburgu" |
| Zn | **REDL** "Redin" >**REDL** "Lollorossa"= **ICEL** "Maugli" = **BUTL** "Cud Voorburgu" >**BUTL** "Zimująca" >**ICEL** "Królowa lata" |

> statistically significantly higher content of compounds or elements (at $p < 0.05$); = differences in content of compounds or elements are statistically insignificant (at $p < 0.05$).

### 3.3. Sugars

Compared with the control, the introduction of I+Se and SA (at each dose) into the nutrient solutions equally contributed to the increases in glucose, fructose, and total sugars (glucose + fructose + sucrose) in all the cultivated varieties of lettuce, except for ICEL "Maugli" (Figure 3). In that cultivar, the levels of all sugars were significantly increased (compared with the control), but only on the media to which SA was introduced at doses of 0.1, 1, and 10 mg·dm$^{-3}$; no such effect was found after the application of I+Se to the nutrient solution without SA. In comparison with the other cultivars, the ICEL cultivar "Maugli" was notable for its high levels of glucose and fructose and its low level of sucrose (Table 1).

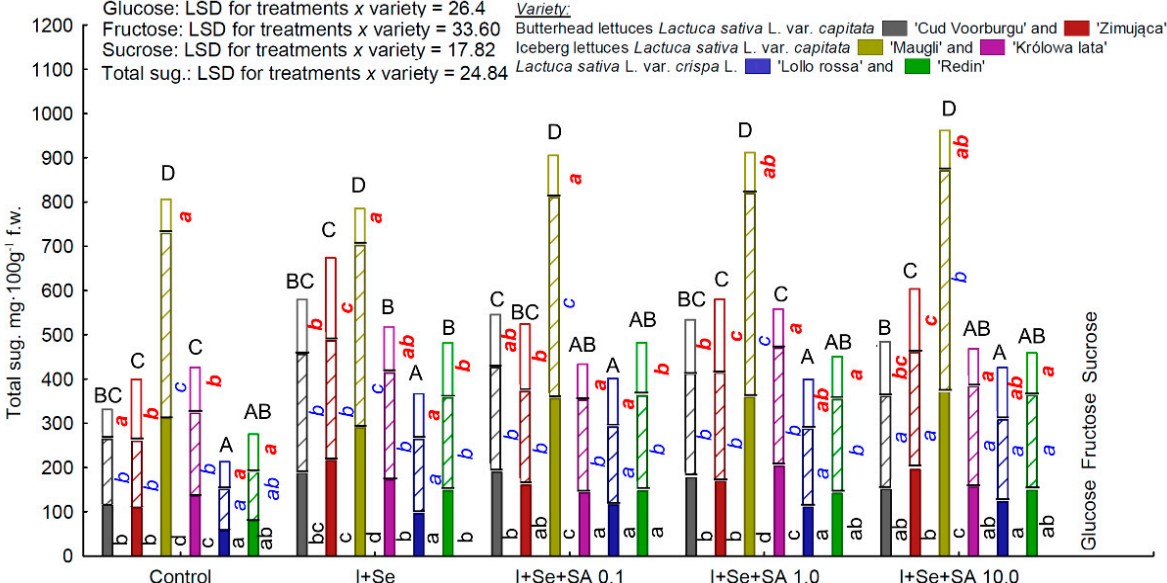

**Figure 3.** Total sugars in lettuce as sum concentrations of glucose (full gradient), fructose (diagonal lines gradient), and sucrose (empty gradient). Means for cultivars followed by the same letters separately for each treatment (type of nutrient solution) and each kind of sugar are not significantly different at *p* < 0.05. Determination of homogeneous groups: lowercase black letters for glucose, lowercase blue italic letters for fructose, lowercase red bold italics for sucrose, and capital black letters for total sugars. LSD for interaction "treatments × variety" at *p* < 0.05. Bars indicate standard errors (*n* = 8). f.w.—fresh matter weight.

### 3.4. Phenols, Phenylpropanoids, Flavonols, and Anthocyanins

The application of I+Se and SA (in each of the three SA doses) to the nutrient solutions caused, to the same extent (compared with the control), significant increases in the concentrations of phenols, phenylpropanoids, flavonols, and anthocyanins in almost all of the lettuce cultivars except ICEL "Królowa lata"; no changes in the levels of these compounds were found in the leaves of this cultivar (Figure 4A–D). In the case of BUTL "Zimująca", only after the application of I+Se+SA 0.1 mg·dm$^{-3}$ were the levels of phenols, phenylpropanoids, flavonols, and anthocyanins in its leaves the same as in those of the control.

The highest concentrations of phenols, phenylpropanoids, flavonols, and anthocyanins were determined in the leaves of BUTL "Zimująca", and the lowest were found in ICEL "Maugli" (Table 1).

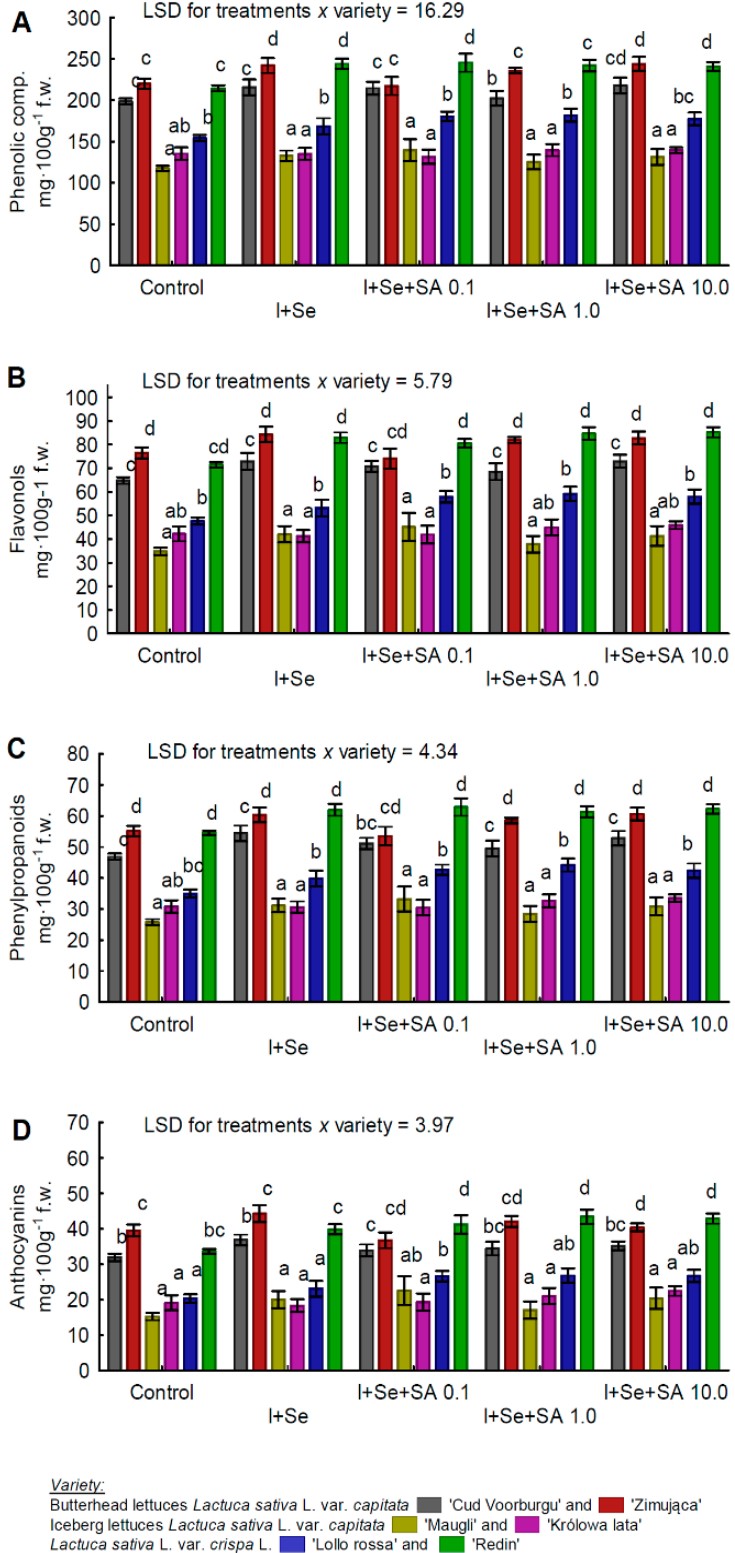

**Figure 4.** Concentrations of phenolic compounds (**A**), flavonols (**B**), phenylpropanoids (**C**), and anthocyanins (**D**) in lettuce. Means for cultivars followed by the same letters separately for each treatment (type of nutrient solution) are not significantly different at $p < 0.05$. LSD for interaction "treatments × variety" at $p < 0.05$. Bars indicate standard errors ($n = 8$). f.w.—fresh matter weight.

### 3.5. Macro- and Micronutrients

The statistical analysis of the results of the macro- and micronutrient contents showed (Figure 5A–G, Figure 6A–F) that the introduction of I+Se and, additionally, SA (at each dose) into the nutrient solutions generally resulted in a significant increase in the Na content of the leaves of all lettuce cultivars. In the case of manganese (Mn), in all lettuce cultivars, only the application of I+Se in combination with SA at a dose of 10 mg·dm$^{-3}$ caused a significant decrease, whereas a dose of 0.1 mg SA dm$^{-3}$ significantly increased the Mn content of the leaves (Figure 6D).

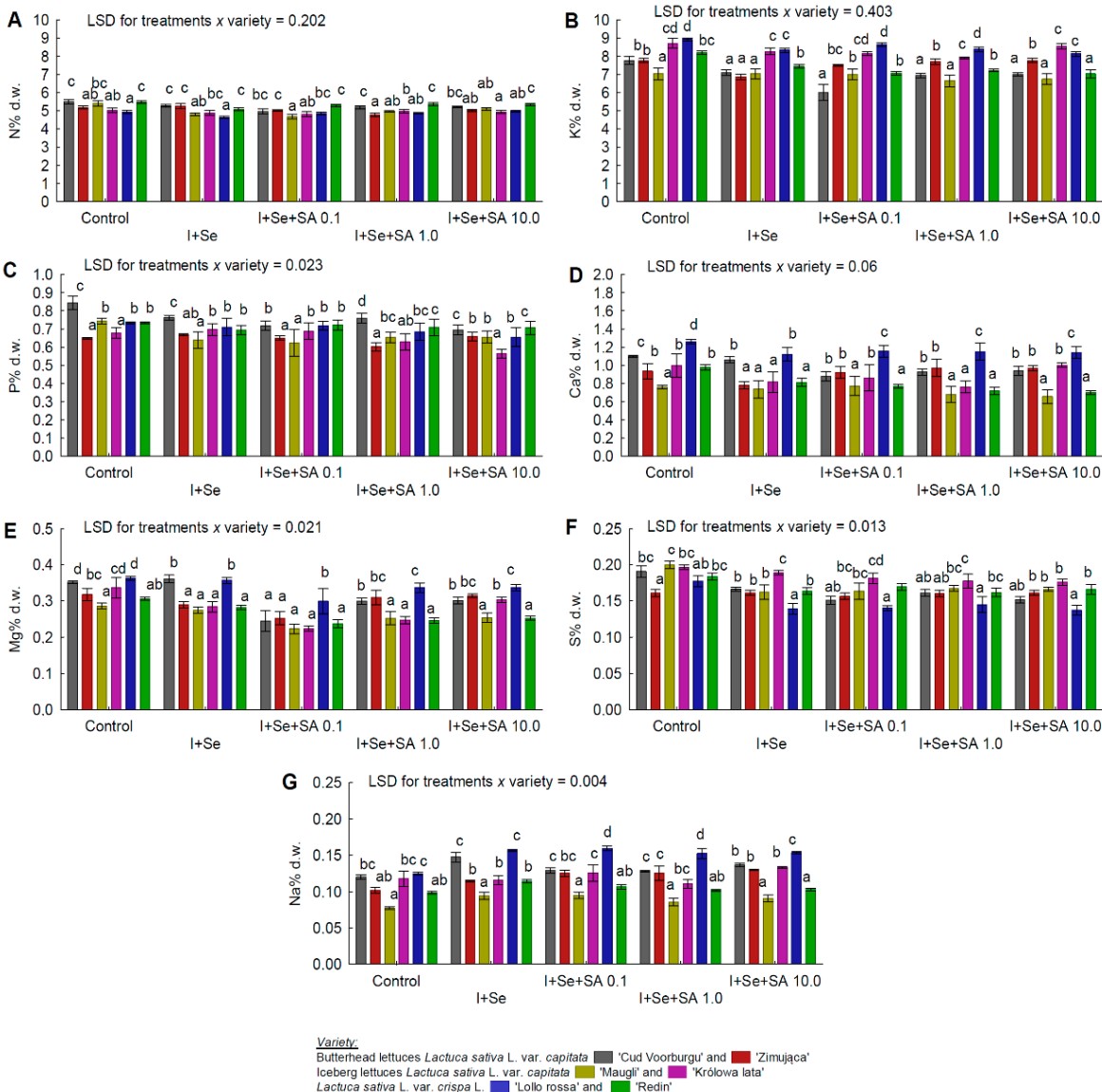

**Figure 5.** Concentrations of macronutrients: N (**A**), K (**B**), P (**C**), Ca (**D**), Mg (**E**), S (**F**), and Na (**G**) in lettuce. Means for cultivars followed by the same letters separately for each treatment (type of nutrient solution) are not significantly different at *p*< 0.05. LSD for interaction "treatments × variety" at *p*< 0.05. Bars indicate standard errors (*n* = 8). d.w.—dry matter weight.

The concentrations of the other nutrients, i.e., N, K, P, Ca, Mg, S, B, Cu, Fe, Mo, and Zn, in the leaves of all lettuce cultivars decreased significantly in all treatments relative to the control. There were approximately a dozen exceptions from the presented general tendency/dependence, which were cultivar-specific traits with respect to the concentration of a particular element. In comparison with the control, some of the exceptions noted included the following:

1.　No effect of SA application on the N content (in REDL "Lollorossa" and "Redin");
2.　No increase in the Na content of the leaves of ICEL "Królowa lata";
3.　No effect of the applied nutrient solutions on the Mo content of the leaves of BUTL "Cud Voorburgu", ICEL "Królowa lata", REDL "Lollorossa", and BUTL "Zimująca";
4.　In BUTL "Cud Voorburgu", a reduction in the Mg content of the leaves but only in the combinations with SA application;
5.　In ICEL "Królowa lata", a significant effect of SA at 10 mg·dm$^{-3}$ on increasing the Ca and B contents, as well as decreasing the P content;
6.　In the leaves of BUTL "Zimująca", a significant increase in the Cu (after application of SA at 0.1 and 10 mg·dm$^{-3}$) and Fe (in all combinations with I, Se, and SA) contents. Also, for BUTL "Zimująca", a significant reduction in the K content only after applying I+Se, a significant reduction in the P content only after applying I+Se+SA 1.0 mg·dm$^{-3}$, and no significant differences in the S content (all media tested) and N content (only I+Se).

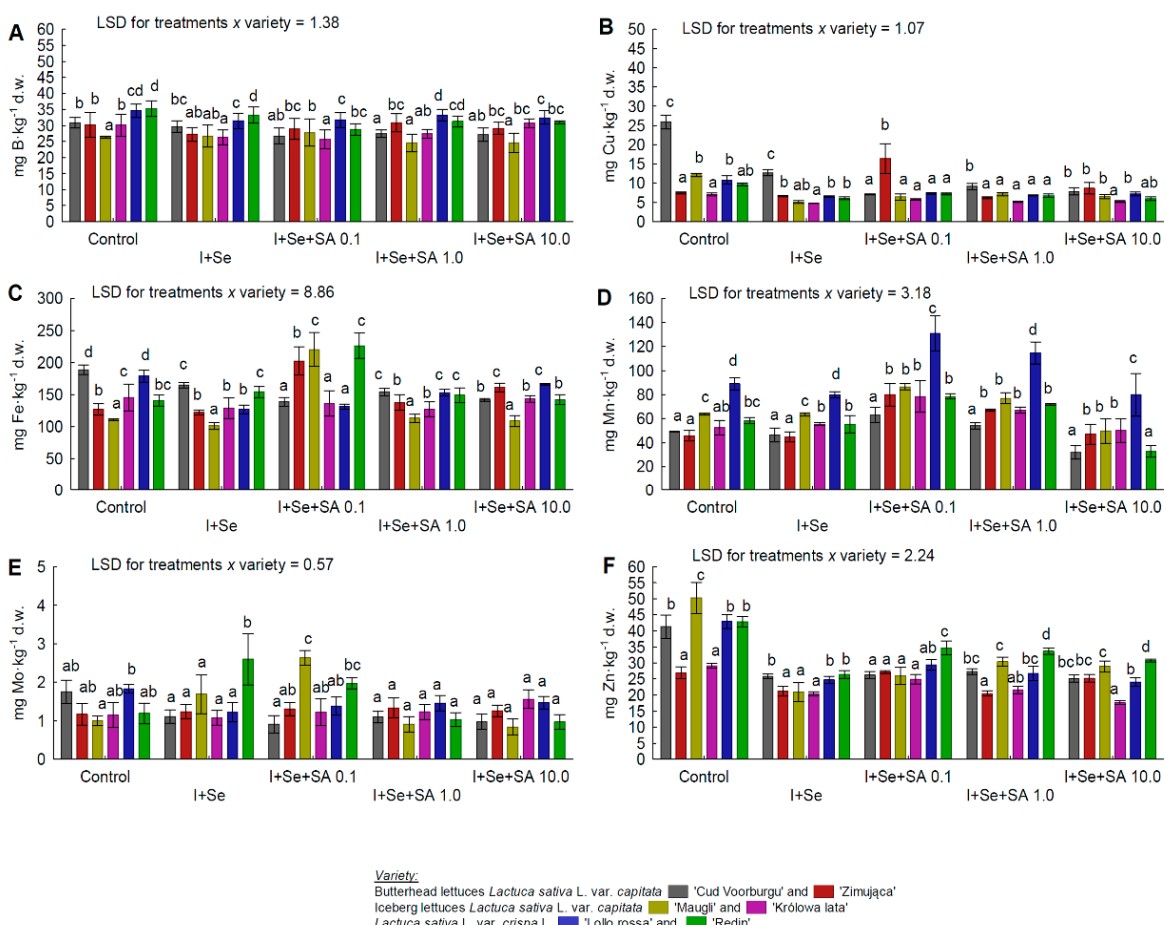

**Figure 6.** Concentrations of micronutrients: B (**A**), Cu (**B**), Fe (**C**), Mn (**D**), Mo (**E**), and Zn (**F**) in lettuce. Means for cultivars followed by the same letters separately for each treatment (type of nutrient solution) are not significantly different at *p*< 0.05. LSD for interaction "treatments × variety" at *p*< 0.05. Bars indicate standard errors (*n* = 8). d.w.—dry matter weight.

Of all the cultivars, REDL "Redin" was characterized by the highest N, Fe, Zn, and Mo contents (like REDL "Lollorossa"), the second-highest P, S, and B contents, a moderate K content, and low Mg, Ca, Na, Cu, and Mn contents (Table 1). In turn, relative to the other cultivars, REDL "Lollorossa" had the highest K, Mg, Ca, Na, B, Mn, and Mo contents, a high Zn content, moderate P, Cu, and Fe contents, and the lowest N and S contents.

## 4. Discussion

Biomass productivity and the chemical composition of lettuce cultivated in soil-less and hydroponic systems depend more on cultivar-specific traits than on the type of growing substrate used for the plants [28]. In our study, lettuce plants were grown in an NFT hydroponic system without any substrate at all. In the case of some of the qualitative traits of lettuce (e.g., ascorbic acid or nitrate content), we demonstrated a different response of each cultivar to the application of I, Se, and SA. For other qualitative traits, we observed a similar response of individual cultivars to the introduction of I, Se, and SA into the nutrient solutions. This was the case, for example, with the concentration of phenolic compounds (Figure 4A–D) or the majority of the mineral elements determined in the plants. Accordingly, it is reasonable to state that the changes in the chemical compositions of individual cultivars observed in the present experiment were the result of the introduction of I, Se, and SA to the nutrient solutions.

### 4.1. TI and Plant Chemical Composition—An Iindicator of Plant Physiological State

Germ et al. [18] found that 4 h of soaking seeds in solutions of $Na_2SeO_3$ with KI or $KIO_3$ caused significant increases in chlorophyll *a* and carotenoids in buckwheat sprouts. However, the same method of treating seeds did not affect the chlorophyll *a* or carotenoid contents of pumpkin sprouts. Only the treatment of seeds with $Na_2SeO_3$+KI had a significant effect on reducing the potential photochemical efficiency of PS II in pumpkin sprouts. Ramos et al. [29], as a result of applying Se as $Na_2SeO_3$ and $Na_2SeO_4$, showed very large differences among 30 lettuce (*Lactuca sativa* L.) varieties in terms of sensitivity to these compounds and in terms of the level of Se accumulation, free amino acids, and the activity of the enzymes GSH-Px (glutathione peroxidase), CAT (catalase), and APX (ascorbate peroxidase). Excessively high doses of either Se or I are harmful, and when toxicity thresholds are exceeded, they are toxic to plants. The I and Se toxicity thresholds for plants depend on the dose and chemical form in which they are applied: $I^-$ and $IO_3^-$ [30,31], as well as $SeO_3^{2-}$ and $SeO_4^{2-}$ [9,10,32,33]. In addition, the sensitivity of plants to I and Se depends on the growing conditions and the method of application. With properly planned agrotechnics, even relatively high doses of these elements can be of little or no harm to plants. For example, in a study by Esringu et al. [33], triple fertigation to the soil with sodium selenite ($Na_2SeO_3$) at doses of 50, 100, and 200 mg·kg$^{-1}$ of soil caused significant increases in yield and growth parameters (head: weight, diameter, and length) of iceberg, romaine, and leaf lettuce. The level of Se accumulation in the three types of botanical varieties of lettuce (iceberg, romaine, and leaf lettuce) varied considerably depending on the dose of $Na_2SeO_3$.

The calculated values of the TI index clearly indicate that the combined application of I+Se (without SA) in the concentrations used in the present study weakened the growth of lettuce plants. However, one can only speak of harmfulness, not of toxicity. We did not observe any symptoms of toxicity on the plants that were characteristic of excessive accumulation of I [30,31] or Se [10,32,33]. The chemical analyses performed in present experiment indirectly indicate that the application of I+Se (without SA) caused changes in the biochemical processes of lettuce plants. Compared with the control, the increases in all the cultivars in the concentrations of phenols, phenylpropanoids, flavonols, anthocyanins, glucose, fructose, sucrose, and total sugars (as well as ascorbic acid in some of them) indicate that the application of I+Se was a stress factor for the lettuce plants. In plants, the increased synthesis of a group of phenolic compounds is a biochemical marker indicating the occurrence of biotic and/or abiotic stress factors [34]. The group of phenolic compounds estimated in the present study (phenols, phenylpropanoids, flavonols, and anthocyanins), synthesized in the processes of secondary plant metabolism, could have been a response to the stress associated with the presence of I and Se. For the synthesis of those compounds, sugars are needed as donors of carbon chains and energy. In our study, the increases in the concentrations of both simple sugars and sucrose (observed after I+Se application) indicate that the plant biochemical processes were directed toward protective responses resulting from the increased accumulation of I and Se. The sugars were, therefore, not used by the plants to grow and develop leaves of lettuce heads (hence, the lowering of the TI index). Instead, they

were collected in the leaves and used for the synthesis of secondary compounds of plant metabolism, including the synthesis of phenolic compounds and ascorbic acid. Ascorbic acid performs antioxidant functions in plant cells; it reacts with many reactive oxygen species (ROS) [35]. This acid is present mainly in the cytoplasm. Ascorbic acid wasalso found to be transported to the apoplast [36,37]. There, it is involved in oxidation–reduction processes that take place under normal conditions, as well as during exposure to abiotic stresses [38].

SA is a compound that is responsible for plant responses to abiotic and biotic stress factors [22]. The results of our study indicate that exogenously applied SA exerted only a very small influence on mitigating the harmful effects to the plants of the application of I+Se to the nutrient solution. In addition, the effect of the SA dose varied depending on the lettuce cultivar. This is reflected in the increased values of TI, as well as in the reduction in the ascorbic acid content; this applies to the relationships of I+Se+0.1 and 1.0 mg·dm$^{-3}$ SA versus I+Se without SA. The finding of only a small, positive effect of SA on relieving stress (associated with a high accumulation of I and Se in plants) in the present experiment may have resulted from the metabolic processes that SA is subjected to in plants. One can mention here the processes of glucosylation, methylation (to methyl salicylate, MeSA), and amino-acid conjugation [39].

The *S*-adenosyl-ʟ-methionine enzyme salicylic acid carboxyl methyltransferase, encoded by the *SAMT* gene, is responsible for the conversion of salicylic acid to MeSA [40,41]. An increase in the level of *SAMT* expression translates into more intense MeSA synthesis, which was noted after the application of exogenous SA in *Atropa belladonna* [42]. The conversion of SA to sugar derivatives, namely 2,3-dihydroxybenzoic acid (2,3-DHBA) and 2,5-dihydroxybenzoic acid (2,5-DHBA), was found in *Arabidopsis* plants [43]. These processes require energy expenditure on the part of plants. The results of our study indicate that each of the cultivars responded, in its own characteristic way, to the exogenous SA with the processes of metabolism and synthesis of both primary (sugars) and secondary (phenolic compounds) compounds of plant metabolism.

### 4.2. Accumulation of Nitrates(V) and Plant Nutritional Status with Respect to Macro- and Micronutrients

As a rule, leaf vegetables accumulate significantly more nitrates(V) than other plant species. The levels of accumulation of nitrates(V) in autumn, winter, and early spring cultivation are much higher than those in late spring and summer [44]. This is because there is less light and, thus, lower photosynthetic activity; the activity of the nitrate reductase (NR) enzyme, which is responsible for the reduction of nitrates(V) in plants, decreases [45,46].

We cultivated lettuce in a greenhouse in autumn, which exposed plants to light deficits. We used supplementary lighting with sodium lamps only between 5:00 and 10:00 a.m. The level of accumulation of nitrates(V) ranged from about 2000 mg $NO_3^-$ kg$^{-1}$fresh weight (f.w.) in the ICEL cultivar "Maugli" to a level no higher than 5000 mg $NO_3^-$kg$^{-1}$f.w. in REDL "Lollorossa". Accordingly, the amounts of nitrates(V) in our lettuce plants did not exceed 5000 mg $NO_3^-$ kg$^{-1}$f.w., i.e., the maximum allowable content for consumers in the European Union [47].

The related body of literature shows that exogenous SA may increase the activity of the NR enzyme, which is responsible for the reduction of nitrates(V) in plants [48]. There were disparate reports on the influence of $SeO_3^{2-}$ (selenate(IV)) on the uptake of $NO_3^-$ ions and the process of their reduction by NR in plants. In a study by Aslam et al. [49], the application of 0.1 mol·m$^{-3}$ $SeO_3^{2-}$ (selenate(IV)) limited the uptake of $NO_3^-$ ions and their reduction by NR in barley seedlings. In the case of a different form of selenium, $SeO_4^{2-}$ (selenate(VI)), this effect on barley seedlings was not produced until a 10-fold higher concentration of $SeO_4^{2-}$, i.e., 1.0 mol·m$^{-3}$,was used. In other studies on wheat seedlings, it was found that $SeO_3^{2-}$, at each of the applied doses of 0.05, 0.15, and 0.45 mmol·kg$^{-1}$ of soil, caused the activation of plant nitrate reductase [50]. There are hypotheses that, in higher plants, the NR enzyme (reducing $NO_3^-$ to $NO_2^-$) may participate in the reduction of $IO_3^-$ to I$^-$. Such a possibility is indicated by the results of research on marine algae [51,52]. The hypothesis is supported by the results of research conducted by Blasco et al. [30] on lettuce. The cited authors showed that

the application of $IO_3^-$ was followed by a significant increase in the nitrate(V) content, which was accompanied by an increase in the activity of the NR enzyme and nitrite reductase (NiR).

In our study, we could have expected that the increased supply of $IO_3^-$ ions to plants would result in a slowdown in the rate of reduction of nitrates(V) and, thus, an increase in the rate of their accumulation in the plants. Meanwhile, in comparison with the control, there was no increase in the level of nitrates(V) in lettuce after using $KIO_3$. Perhaps this resulted from the application of this compound together with $Na_2SeO_3$. What is important, however, is that the combined application of I+Se caused, in individual cultivars, a reduction in the concentration of nitrates(V) or had no effect on their levels in lettuce. In terms of the impact of SA on the level of accumulation of nitrates(V), the response of individual cultivars varied. Advantageously, from the consumer's point of view, there were reductions (as a result of the application of I, Se, and SA) in the nitrate(V) content of the leaves of BUTL "Cud Voorburgu" and "Zimująca", REDL "Redin", and, to some extent, ICEL "Królowa lata". This, admittedly, translated into a reduction in the total level of N in the leaves of those lettuce cultivars; however, the degree of their nitrogen nutrition was still close to the range considered to be optimal for lettuce, which is 3.3–4.8 N% d.w. [53–55].

In the study by Blasco et al. [55], the application of $IO_3^-$ at doses of 20, 40, and 80 μM (compared with the control) caused a significant increase in the weight of romaine lettuce heads grown in pearlite. The important finding of that study was that after the application of $IO_3^-$ (as opposed to the use of $I^-$), the lettuce plants were optimally nourished with respect to all macro- and micronutrients. In the case of Mg and Fe, the authors even found an improvement in the nutritional status of the plants in relation to these two elements.

In our study, the tested applications of I, Se, and SA caused, with a few exceptions described in detail in Section 3.4, significant reductions (relative to the control) in the concentrations of N, K, P, Ca, Mg, S, B, Cu, Fe, Mo, and Zn in the leaves of all lettuce cultivars. Nevertheless, the changes in the mineral composition of lettuce shown in the present experiment should not negatively affect the plants. We came to this conclusion after a detailed analysis of literature data regarding the optimal mineral composition of lettuce [53–55]. The fluctuations in the concentrations of N, K, P, Ca, Mg, S, B, Cu, Fe, Mn, Mo, and Zn recorded in all the cultivars did not exceed (downward or upward) the lower or upper limits of optimum values of these elements for lettuce [52–54]. It should be noted that the sulfur nutritional status of all cultivars grown on each type of nutrient solution, including the control, was below the lower threshold of the optimal S content of lettuce, which is about 0.2–0.25% S d.w. [53–55]. There are data indicating that lettuce gives a satisfactory yield, even at an S content of the plants of around 0.05–0.1%S d.w. [56]. This is due to the fact that lettuce is among the species with low nutritional needs for S. In our study, the S content of the individual lettuce cultivars was not lower than 0.14% S d.w. For this reason, the indicated nutritional status with respect to S did not represent a deficit for the lettuce plants.

In a study by Koudela and Petřiková [57], the yield potential of five cultivars of *Lactuca sativa* L. var. *crispa* L. (including "Lollorossa" and "Redin") depended on the climatic conditions and the time of cultivation (spring, summer or autumn). The authors showed that the K, Na, Mg, and $NO_3^-$ contents of lettuce were variable features in both "Lollorossa" and "Redin" in the three growing seasons, with "Redin" always forming heads with a significantly greater weight than in "Lollorossa". In the case of the cultivar "Redin", we alsoproved that it had the highest yield potential, expressed by head weight; detailed yield results were included in an earlier publication [58]. At the same time, the cultivar "Redin" was characterized by the greatest sensitivity to the application of I+Se and SA at the three doses, which is evidenced by the value of the TI index. In addition, similarly to Koudela and Petřiková [57], we showed that the cultivar "Lollorossa" was characterized by a higher Ca content than "Redin".

The Na content (Na derived from fertilizers) determined in the present study in the control medium was about 13.5 mg Na $dm^{-3}$. The significant increase, relative to the control, in the Na content of the leaves of all lettuce cultivars (after the application of I+Se and, additionally, SA) was most

probably associated with the additional introduction of selenium in the form of $Na_2SeO_3$ into the nutrient solutions. Apart from that, the nutrient solutions were also characterized by a high K content (170 mg·dm$^{-3}$). The amount of K that was introduced into the media with $KIO_3$ was too small to be able to effectively increase the K supply for the plants. We conclude, therefore, that the effect of the reduced K concentration in the leaves observed in the individual lettuce cultivars relative to the control was the response of the plants to the application of I, Se, and SA.

Ramos et al. [29] demonstrated a synergistic effect of $Na_2SeO_4$ application on the level of sulfur accumulation in lettuce. It depended on the expression and assimilation of the genes involved in Se/S uptake and on the genotypes of the lettuce cultivars tested by those authors. In our study, we used $Na_2SeO_3$ and not $Na_2SeO_4$. Previous studies by Smoleń et al. [58] proved that the introduction of $Na_2SeO_4$ and $KIO_3$ into the nutrient solutions did not affect the concentrations of P and S in the leaves and roots of lettuce plants cultivated in an NFT hydroponic system.

In plants, $SeO_3^{2-}$ ions are transported by the same protein transporters as phosphates, whereas $SeO_4^{2-}$ ions are transported by sulfate transporters [59]. In our study, the application of $SeO_3^{-}$ (together with $IO_3^{-}$, without SA) caused a significant decrease in the P content of the leaves in the following cultivars: BUTL "Cud Voorburgu", ICEL "Maugli", REDL "Lollorossa", and "Redin". At the same time, in the same cultivars, a decreased S content was found. It should be mentioned that the presence of I+Se ($SeO_3^{-} + IO_3^{-}$) in the nutrient solution caused a significant increase, relative to the control, in the P content of the roots of all six cultivars, while it did not affect the S content (data not shown). These results indicate the inhibitory effect of the simultaneous application of I+Se ($SeO_3^{-} + IO_3^{-}$) on P transport from the roots to leaves in those four cultivars; this effect was not observed in BUTL "Zimująca" or ICEL "Królowa lata".

## 5. Conclusions

The study results show that the combined application of I+Se (without SA) is a mild abiotic stress factor for the four tested lettuce cultivars. This is indicated by the results of the calculated TI index for ICEL "Maugli" and "Królowa lata", as well as REDL "Lollorossa" and "Redin". In contrast, the simultaneous application of I+Se was not harmful to two cultivars: BUTL "Cud Voorburgu" and "Zimująca". Based on the results for TI, it can be concluded that the introduction of exogenous SA into the nutrient solution (only at 1.0 mg SAdm$^{-3}$ of nutrient solution) exerted a biostimulatory effect on the plants of all six cultivated varieties of lettuce. These results provide the basis for concluding that 1.0 mg SAdm$^{-3}$ has an anti-stress effect on plants; it reduces the stress-inducing effect of combined I+Se application on lettuce plants. In this study, this was manifested by an improvement in biomass productivity (expressed by TI) of all lettuce cultivars in relation to the cultivation in the nutrient solution containing I+Se without SA application.

The determined parameters of the chemical compositions of the heads of all lettuce cultivars provide evidence that the treatment with I+Se activated metabolic pathways in the plants associated with the plant response to stress due to the increased accumulation of these elements. The exogenously supplied SA (despite improving the TI index) had a relatively small impact on the determined parameters of the chemical compositions of the lettuce plants. At the biochemical level, this indicates that there was little or no effect of SA on mitigating the stress of I+Se application. This could have resulted from the difficulty in the detection—with the standard analyses used in this study—of the metabolism/conversion processes that exogenous SA is subjected to in plants.

In spite of everything, in our opinion, the combined biofortification of plants with I and Se with the use of SA should be performed. Future studies should only focus on refining less stressful doses of I, Se, and SA for plants. We come to this conclusion because the tested combinations I+Se and I+Se+SA resulted, for the majority of lettuce cultivars, in beneficial (to the consumer) changes in the chemical compositions of the lettuce plants. It should be pointed out here that the nitrate(V) content was reduced, and the levels of sugars, phenols, phenylpropanoids, flavonols, and anthocyanins were

increased. In addition, it is worth emphasizing the increase in the ascorbic acid content of the leaves, which only occurred in BUTL "Cud Voorburgu" and "Zimująca", and REDL "Redin".

A separate issue is the impacts of I, Se, and SA on plant mineral balance. Relative to the control, the use of I, Se, and SA had significant upward or downward influences on the concentrations of N, K, P, Ca, Mg, Na, B, Cu, Fe, Mn, Mo, and Zn in lettuce. However, it should be emphasized that, regardless of everything, the concentrations of these elements in all the cultivars remained within the ranges considered to be optimal for lettuce.

**Author Contributions:** Methodology, S.S., I.K., W.S., and P.K.; formal analysis, S.S.; funding acquisition, S.S. and I.K.; investigation, I.K., M.G., and W.S.; resources, S.S. and I.K.; supervision, S.S.; visualization, S.S., I.K., and M.G.; writing—original draft, S.S. and I.K.; writing—review and editing, S.S., I.K., P.K., and U.B.K.

**Funding:** This research was financed by the Ministry of Science and Higher Education of the Republic of Poland.

**Acknowledgments:** The application of salicylic acids with iodine into plants was taken from our patented solution: Sady W, Smoleń S, Ledwożyw-Smoleń S., Methods of biofortification of vegetables with iodine. Patent No. P.410808, Polish Patent Office 07.11.2017.

**Conflicts of Interest:** The authors declare that the research was conducted in the absence of any commercial or financial relationships that could be construed as potential conflicts of interest.

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
