# Peer review of "Changes in the Chemical Composition of Six Lettuce Cultivars (Lactuca sativa L.) in Response to Biofortification with Iodine and Selenium Combined with Salicylic Acid Application"

_agronomy, doi:10.3390/agronomy9100660_

Round 1

Reviewer 1 Report

Major Comments:

All Figures of the manuscript are “blind”. It is extremely difficult to read the information.

-Thus, Fig. 1 may be improved by elevating the X-axes to 40-50 value of “index of tolerance” In this case differences between the columns will be more visual.

       - Fig.3: the values of glucose, fructose and sucrose columns are extremely small. May be it will be better to combine the data with the total sugar content using complex columns that will allow to better see  the differences of glucose, fructose and sucrose distribution in the total amount of sugar. In this case, it will be possible to replace 4 figures by one with higher resolution

        -Fig.4: utilization of similar ordinate scale for total phenolics, flavonols, anthocyanins, phenylpropanoids results in lack of possibility of seeing any differences in the latter three cases. May be it will be better to use maximum value for flavanols and phenylpropanoids- 100, and for anthocyanins-50. Think that everybody knows that all these compounds belong to polyphenols

        -Fig.5- the same as for Fig.1

2) The phrase: ” All the cultivars of lettuce were also analysed for the following: I, Se, selenomethionine (SeMet),  selenocysteine (SeCys), SA and proline” (page 5) gives practically no information though explains lack of certain data in the manuscript.  If you begin speaking about selenium and iodine accumulation levels then give briefly the results (just one or two sentances), otherwise it has no sense.

3) Antioxidant activity data are desirable

4) The authors do not discuss the correlations between the parameters investigated

5) Can’t understand how sugar content was determined if the authors used ethanolic extracts: sucrose, glucose and fructose are not soluble in ethanol

6) All the results are given per fresh weight of plants – was there no difference in dry matter content of all cultivars? If there were some then the comparison between the treatments is not precise

Minor comments:

- the word “leaves” on all Figures should be deleted- because it is the only object of investigation

-a misprint on Page 5: “analysed” instead of “analyzed”

Author Response

Dear Editor and Reviewer No. 1

First of all, we would like to thank you for your time and effort put into the review as well as a given opportunity to improve our manuscript.

We hope that all the changes and improvements will meet your requirements and acceptance. We declare willingness to make all further modifications if that is the decision of the Editor and all Reviewers.

All corrections and introductions in the manuscript are marked with blue font. The proof reading of the article was conducted by MDPI English editing - service (https://www.mdpi.com/authors/english) that is service recommended by Agronomy journal (ISSN 2073-4395)

Below you will find a point-by-point response to the reviewer’ comments

In the review:

["Major Comments: All Figures of the manuscript are “blind”. It is extremely difficult to read the information.

-Thus, Fig. 1 may be improved by elevating the X-axes to 40-50 value of “index of tolerance” In this case differences between the columns will be more visual.

   - Fig.3: the values of glucose, fructose and sucrose columns are extremely small. May be it will be better to combine the data with the total sugar content using complex columns that will allow to better see  the differences of glucose, fructose and sucrose distribution in the total amount of sugar. In this case, it will be possible to replace 4 figures by one with higher resolution

        -Fig.4: utilization of similar ordinate scale for total phenolics, flavonols, anthocyanins, phenylpropanoids results in lack of possibility of seeing any differences in the latter three cases. May be it will be better to use maximum value for flavanols and phenylpropanoids- 100, and for anthocyanins-50. Think that everybody knows that all these compounds belong to polyphenol./.ls

        -Fig.5- the same as for Fig.1"]

Our answer:

All figures have been modified according to your suggestions.

Reviewer No. 2 also asked for the figures to be corrected - in accordance with his/her comments we introduced letter designations for homogeneous groups

In the review:

"2) The phrase: ” All the cultivars of lettuce were also analysed for the following: I, Se, selenomethionine (SeMet),  selenocysteine (SeCys), SA and proline” (page 5) gives practically no information though explains lack of certain data in the manuscript.  If you begin speaking about selenium and iodine accumulation levels then give briefly the results (just one or two sentances), otherwise it has no sense."

Our answer:

This fragment has been re-edited. We only mentioned analyzing the content: I, Se, SeMet and SeCys. Only the necessary data were quoted. This fragment now contains short information.

In the review:

"3) Antioxidant activity data are desirable"

Our answer:

Please be advised that we did not perform the analysis of antioxidant activity of lettuce plants in that project. Unfortunately, we cannot conduct these additional analyzes as the plant material was not protected in the appropriate manner. After receiving the review, we considered repeating the vegetation experiment as to perform these analyzes, yet in the current time there is no required amount of place in the greenhouse due to the implementation of numerous other research projects. We kindly ask you to accept our publication without the results of antioxidant activity analyzes even though we are aware that this aspect is very important as iodine can modify antioxidant activity. However, for objective reasons, we cannot enrich our publication with these results.

In the review:

"4) The authors do not discuss the correlations between the parameters investigated."

Our answer:

In our publication we presented 23 different parameters of the chemical composition of lettuce. Therefore, when calculating the correlation matrix we obtained 529 correlation coefficients (23 x 23 = 529). This is the amount of data common to all varieties and treatments (types of nutrient solution - without taking into account the separate reaction of 6 varieties). However, when calculating correlation coefficients, this would have to be done separately for each variety: 6 x 23 x 23 = 3 174 correlation coefficients.

While writing the discussion, we tried to highlight the physiological and biochemical aspects of the effects of I, Se and SA on the chemical composition of lettuce, which was in line with the research aim.

Dear Reviewer and Editor - please accept the publication in its current form. Additional consideration of the correlation problem between parameters investigated was very difficult due to the large amount of numerical data. This would require a significant extension of the discussion. The most sensible approach would be to calculate the correlation between the iodine and selenium content and the other parameters for each lettuce variety separately. We declare that we can supplement the publication with additional extensive descriptions of the correlation matrix, but we are not convinced of its cognitive value.

In the review:

"5. Can’t understand how sugar content was determined if the authors used ethanolic extracts: sucrose, glucose and fructose are not soluble in ethanol."

Our answer:

We would like to kindly inform you that ethanolic extracts can be easily used for sugar determination. This is possible because sugars are already dissolved in plant tissues. When it comes to making the standard curve (in instrumental analysis, e.g. HPLC, CE or UV-VIS), the concentrated stock solution is made on water alone or about 75% ethanol. Further working patterns can already be made on a matrix of concentrated ethyl alcohol. Plant extraction with ethanol was also proposed in the standard procedure for sugar analysis in plant tissues by Yemm, E. W., & Willis, A. (1954). The estimation of carbohydrates in plant extracts by anthrone. Biochemical journal, 57(3), 508 as well as other procedures, e.g. Buysse, J. A. N., & Merckx, R. (1993). An improved colorimetric method to quantify sugar content of plant tissue. Journal of Experimental Botany, 44(10), 1627-1629; Chow, P. S., & Landhäusser, S. M. (2004). A method for routine measurements of total sugar and starch content in woody plant tissues. Tree physiology, 24(10), 1129-1136.

In the review:

"6) All the results are given per fresh weight of plants – was there no difference in dry matter content of all cultivars? If there were some then the comparison between the treatments is not precise."

We would like to inform that I+Se fertilization and additional SA application did not have a statistically significant effect on the dry matter content in the leaves of individual lettuce varieties. Therefore, these results were omitted in our publication.

In addition, only a part of our results are presented on a fresh weight basis [(L-ascorbic acid, sugars, phenols, phenylpropanoids, flavonols, anthocyanins and nitrates(V)]. Other results are presented on a dry weight basis (content of N by the Kjeldahl method, and K, P, Ca, Mg, S, Na, B, Cu, Fe, Mn, Mo and Zn) - see also sections 2.2.1 and 2.2.2. Such a division results not only from the analytical methods used but also from the standards adopted in the literature for presenting the results of these analyzes. As a rule, nutrition tables for the content of: L-ascorbic acid, sugars, phenols, phenylpropanoids, flavonols, anthocyanins and nitrates(V) contain values on a fresh weight basis. If we converted the content of these substances to the dry weight of lettuce, it would be difficult to compare our results those presented by other authors. In view of the above, please accept the method of result presentation in its current form.

In the review:

"[Minor comments:

- the word “leaves” on all Figures should be deleted- because it is the only object of investigation

-a misprint on Page 5: “analysed” instead of “analyzed"]"

 Our answer:

All corrected as suggested. Thank you. Indeed, we have unnecessarily entered the word "Leaves" in all figures.

We hope that all the changes and improvements will meet your requirements and acceptance.

We inform that the manuscript has not been submitted for publication elsewhere.
All co-authors have contributed to this article and all agree to submit it into the Agronomy journal (ISSN 2073-4395).

There are no conflicts of interests. We would be grateful for the acceptance of our publication.

Yours sincerely,

On behalf of the Authors Team,

Dr.Sc. Sylwester Smoleń, Associate professor

Unit of Plant Nutrition

Department of Plant Biology and Biotechnology
Faculty of Biotechnology and Horticulture, University of Agriculture in Krakow

Al 29 Listopada 54

31-425 Kraków

POLAND

sylwester.smolen@urk.edu.pl   Sylwester.Smolen@interia.pl

Reviewer 2 Report

Changes in the Chemical Composition of Six Lettuce Cultivars (Lactuca sativa L.) in Response to Biofortification with Iodine and Selenium Combined with Salicylic Acid Application

Biofortification and mineral composition of agriculturally important plants is a highly important area of research.The article discusses the biofortification of different lettuce cultivars with iodine and selenium in combination with salicylic acid.

The introduction is consistent with a clearly defined problem. The experiment is adequately described with sufficient data.In results some modifications in Figures needs to be done. The authors used the relevant literature to discuss their results.

L 120: Abbreviation “ICP-OES” is first mentioned so it should be explained.

L 138: Abbreviation “FIA” – explain

Figures 1,2,3,4,5 and 6:

1) In figure captions it is written “ Means followed by the same letters are not..... But I can’t see any letters in figures 1,2,3,4,5 and 6. You should add the letters indicating the statistical difference between treatments.

2) Maybe you could use different colors for different treatments. It might be more appropriate since in some figures it is very difficult to distinguish between different cultivars.

L 289, 311, 314, 320, 339, 401, 424: ...”by us”.... It would be more appropriate to use for example “in present experiment or present study ...”

Author Response

Dear Editor and Reviewer No. 2

First of all, we would like to thank you for your time and effort put into the review as well as a given opportunity to improve our manuscript.

We hope that all the changes and improvements will meet your requirements and acceptance. We declare willingness to make all further modifications if that is the decision of the Editor and all Reviewers.

All corrections and introductions in the manuscript are marked with blue font. The proof reading of the article was conducted by MDPI English editing - service (https://www.mdpi.com/authors/english) that is service recommended by Agronomy journal (ISSN 2073-4395).

Below you will find a point-by-point response to the reviewer’ comments

In the review:

"Biofortification and mineral composition of agriculturally important plants is a highly important area of research.The article discusses the biofortification of different lettuce cultivars with iodine and selenium in combination with salicylic acid.

The introduction is consistent with a clearly defined problem. The experiment is adequately described with sufficient data. In results some modifications in Figures needs to be done. The authors used the relevant literature to discuss their results.

cases. May be it will be better to use maximum value for flavanols and phenylpropanoids- 100, and for anthocyanins-50. Think that everybody knows that all these compounds belong to polyphenols.    -Fig.5- the same as for Fig.1""

 Our answer:

Dear Reviewer. Thank you for your critical comments. We inform you that all figures have been corrected in accordance with your and the reviewer's No. 1 suggestions.

In the review:

"[L 120: Abbreviation “ICP-OES” is first mentioned so it should be explained.

L 138: Abbreviation “FIA” – explain

Figures 1,2,3,4,5 and 6:

1) In figure captions it is written “ Means followed by the same letters are not..... But I can’t see any letters in figures 1,2,3,4,5 and 6. You should add the letters indicating the statistical difference between treatments.

2) Maybe you could use different colors for different treatments. It might be more appropriate since in some figures it is very difficult to distinguish between different cultivars.

L 289, 311, 314, 320, 339, 401, 424: ...”by us”.... It would be more appropriate to use for example “in present experiment or present study ...]”

Our answer:

All corrected as suggested. We introduced the letter designations of homogeneous groups in the figures.

We hope that all the changes and improvements will meet your requirements and acceptance.

We inform that the manuscript has not been submitted for publication elsewhere.
All co-authors have contributed to this article and all agree to submit it into the Agronomy journal (ISSN 2073-4395).

There are no conflicts of interests. We would be grateful for the acceptance of our publication.

Yours sincerely,

On behalf of the Authors Team,

Dr.Sc. Sylwester Smoleń, Associate professor

Unit of Plant Nutrition

Department of Plant Biology and Biotechnology
Faculty of Biotechnology and Horticulture, University of Agriculture in Krakow

Al 29 Listopada 54

31-425 Kraków

POLAND

sylwester.smolen@urk.edu.pl   Sylwester.Smolen@interia.pl

Round 2

Reviewer 1 Report

Thank you for your work. The new version seems to be much better